# Polymorphisms in intron 1 of HLA-DRA differentially associate with type 1 diabetes and celiac disease and implicate involvement of complement system genes C4A and C4B

Ozkan Aydemir[1]*, Jeffrey A Bailey[2], Daniel Agardh[3], Åke Lernmark[3], Janelle A Noble[4], Agnes Andersson Svärd[3], Elizabeth P Blankenhorn[5], Hemang M Parikh[6], Anette-G Ziegler[7], Jorma Toppari[8,9], Beena Akolkar[10], William A Hagopian[11,12], Marian J Rewers[13], John P Mordes[1], TEDDY Study Group

[1]University of Massachusetts Chan Medical School, Worcester, United States; [2]Department of Pathology and Laboratory Medicine and Center for Computational Molecular Biology, Brown University, Providence, United States; [3]Department of Clinical Sciences, Lund University CRC Skane University Hospital, Malmö, Sweden; [4]University of California, San Francisco, San Francisco, United States; [5]Drexel University College of Medicine, Philadelphia, United States; [6]Health Informatics Institute, Morsani College of Medicine, University of South Florida, Tampa, United States; [7]Institute of Diabetes Research, Helmholtz Zentrum München, Klinikum rechts der Isar, Technische Universität München, and Forschergruppe Diabetes e.V., Neuherberg, Germany; [8]Research Centre for Integrative Physiology and Pharmacology, Institute of Biomedicine, and Centre for Population Health Research, University of Turku, Turku, Finland; [9]Department of Pediatrics, Turku University Hospital, Turku, Finland; [10]National Institute of Diabetes and Digestive and Kidney Diseases, Bethesda, United States; [11]Pediatrics, Indiana University School of Medicine, Indianapolis, United States; [12]Department of Medicine, University of Washington, Seattle, United States; [13]Barbara Davis Center for Childhood Diabetes, University of Colorado, Aurora, United States

*For correspondence:
ozkanaydemir@gmail.com

## eLife assessment

This study presents **valuable** findings on genetic risk factors for type 1 diabetes and celiac disease using a large cohort from the Environmental Determinants of Diabetes in the Young (TEDDY) study. The evidence supporting the claims of the authors is **solid**, although the inclusion of the genetic effect of this locus on individuals with different genetic backgrounds would have strengthened the study. The work will be of interest to population geneticists working on diabetes and celiac disease.

**Abstract** Polymorphisms in genes in the human leukocyte antigen (HLA) class II region comprise the most important inherited risk factors for many autoimmune diseases, including type 1 diabetes (T1D) and celiac disease (CD): both diseases are positively associated with the HLA-DR3 haplotype (*DRB1\*03:01-DQA1\*05:01-DQB1\*02:01*). Studies of two different populations have recently documented that T1D susceptibility in HLA-DR3 homozygous individuals is stratified by a haplotype

consisting of three single nucleotide polymorphisms ('tri-SNP') in intron 1 of the *HLA-DRA* gene. In this study, we use a large cohort from the longitudinal 'The Environmental Determinants of Diabetes in the Young' (TEDDY) study to further refine the tri-SNP association with T1D and with autoantibody-defined T1D endotypes. We found that the tri-SNP association is primarily in subjects whose first-appearing T1D autoantibody is to insulin. In addition, we discovered that the tri-SNP is also associated with CD, and that the particular tri-SNP haplotype ('101') that is negatively associated with T1D risk is positively associated with risk for CD. The opposite effect of the tri-SNP haplotype on two DR3-associated diseases can enhance and refine current models of disease prediction based on genetic risk. Finally, we investigated possible functional differences between the individuals carrying high and low-risk tri-SNP haplotypes and found that differences in complement system genes C4A and C4B may underlie the observed divergence in disease risk.

## Introduction

Type 1 diabetes (T1D) is an autoimmune disease that results from the destruction of pancreatic islet β cells by autoreactive T cells (*Wang et al., 2017*). Clinical onset is preceded by a prodrome of β-cell autoimmunity, usually marked by the appearance of autoantibodies directed principally at insulin (IAA), glutamic acid decarboxylase (GADA), and islet antigen-2 (IA-2A). T1D is a non-Mendelian polygenic disorder involving the interaction of multiple gene variants, environmental factors, and immunoregulatory dysfunction (*Wang et al., 2017*). Only about 12% of persons with T1D have a first-degree relative (FDR) with the disease, a percentage that has remained unchanged despite increasing incidence (*Dahlquist et al., 1989*; *Parkkola et al., 2013*). The environmental factors that may precipitate disease in genetically susceptible individuals and the mechanisms by which they act have not been completely identified.

Gluten enteropathy, or CD, is also an autoimmune disorder. It results in atrophy of intestinal villi in the context of intestinal T cell inflammation and is associated with the presence of transglutaminase autoantibodies (tTGA) (*Husby et al., 2012*; *Leonard et al., 2017*). Although gluten is necessary for the disease, it is not by itself sufficient, and CD is likely triggered by other environmental exposures (*Andrén Aronsson et al., 2019*). Clinical disease is characterized by a range of symptoms and signs, including abdominal discomfort, disordered intestinal motility, and nutrient malabsorption. Like T1D, CD is a non-Mendelian polygenic disorder involving the interaction of multiple gene variants, environmental factors, and immunoregulatory dysfunction. On average, about 7.5% of persons with CD have a first-degree relative with the disorder, but the prevalence varies with the relationship, sex, and geographic location (*Singh et al., 2015*).

The major genetic loci for both T1D and CD susceptibility encode major histocompatibility complex (MHC) class II cell surface antigens; they are encoded by the HLA complex genes on chromosome 6 (*Gutierrez-Arcelus et al., 2016*). Together with the T cell receptor (TCR) and an autoantigenic peptide, these glycoproteins form the 'tri-molecular complex' that is central to cell-mediated immune and autoimmune responses.

T1D susceptibility is most strongly associated with polymorphisms in HLA class II genes. The major HLA class II genes that predispose to T1D are *HLA-DRB1*, *DQA1*, and *DQB1*, which encode the HLA DR and DQ heterodimers (*Polychronakos and Li, 2011*). Most T1D patients bear genetic haplotypes known as HLA-DR3 (*DRB1\*03:01-DQA1\*05:01-DQB1\*02:01*) and HLA-DR4 (*DRB1\*04:01-, \*04:02-, \*04:04-, or \*04:05-DQA1\*03:01-DQB1\*03:02*). The DR3 haplotype can also be abbreviated as DR3-DQ2, with DR3 referring to the presence of the *DRB1\*03:01* gene and DQ2 referring to the heterodimeric protein product of the *DQA1\*05:01* and *DQB1\*02:01* genes. The DR4 haplotype represents any of the *DRB1\*04*-encoded allotypes with DQ8, the heterodimeric protein product of the *HLA-DQA1\*03:01* and *DQB1\*03:02* genes (DR4-DQ8). T1D is also associated with HLA class I genes (*Noble et al., 2010*; *Polychronakos and Li, 2011*) and with class II *HLA-DPB1* genes (*Bradfield et al., 2011*). Genome-wide association studies (GWAS) revealed >50 non-MHC loci associated with the disease (*Bradfield et al., 2011*) and more recent analyses have extended that number to >75 (*Robertson et al., 2021*). Because their individual contribution to total risk is small (*Baranzini, 2009*), the analysis of non-MHC risk loci has not improved our ability to predict disease based on genotype alone.

In CD, about 90% of patients carry DR3-DQ2, and most of the remaining CD patients carry the DR4-DQ8 haplotype (*Brown et al., 2019*). Fine mapping of the HLA region in persons with CD has identified additional associations that account for approximately 18% of the genetic risk (*Gutierrez-Achury et al., 2015*). These loci, together with the >50 known non-MHC susceptibility loci identified by GWAS, are thought to explain up to 48% of overall CD heritability (*Gutierrez-Achury et al., 2015*).

The DQ2 and DQ8 haplotypes, positively associated with CD, are carried on the DR3-DQ2 and DR4-DQ8 haplotypes, respectively, that are predisposing for T1D. The occasional presence of both T1D and CD in the same individual is, therefore, not surprising. The rate at which CD occurs in persons with T1D varies substantially among various populations and reportedly ranges from 2.5 to 16.4% (*De Vitis et al., 1996*; *Szaflarska-Popławska, 2014*). In the HLA-selected TEDDY study, T1D autoimmunity usually preceded CD autoimmunity in cases that had both (*Hagopian et al., 2017*).

In the case of T1D, HLA genotyping can identify persons at risk for the disease, but only about 1 in 15 (~7%) individuals with even the highest-risk HLA genotype identified in Europeans (HLA-DR3/DR4 heterozygotes) develop the disorder (*Rewers et al., 1996*). Similarly, only about 3% of persons with DQ2 or DQ8 develop celiac disease (*Brown et al., 2019*). Recent studies that coupled genetic analysis with phenotypic markers have improved our ability to predict islet autoantibodies and T1D in the general population (*Bonifacio et al., 2018*), but additional genetic risk remains to be discovered (*Pierce et al., 2013*). Understanding of the detailed genetic risk conferred by HLA-DR and -DQ-encoding genes remains incomplete and provided the impetus for these studies.

We have previously reported that susceptibility to T1D among persons homozygous for the high-risk DR3-DQ2 haplotype is powerfully modulated by variation within intron-1 of the *HLA-DRA* gene (*Aydemir et al., 2019*), a gene that, unlike most HLA-encoding genes, is considered essentially invariant and largely ignored in HLA association studies. This variation comprises three single-nucleotide polymorphisms (SNPs): rs3135394, rs9268645, and rs3129877 (referred to here as the 'tri-SNP') in a 100 bp interval within intron 1. We called the T1D risk haplotype (nucleotides AGG) '010' using (0) to designate the reference and (1) alternate alleles, respectively. T1D protection was conferred by the '101' haplotype (GCA). The effect of the tri-SNP on T1D susceptibility was discovered in an analysis of samples drawn from the Type 1 Diabetes Genetics Consortium (T1DGC), the

**Table 1.** Distribution of human leukocyte antigen (HLA) and Tri-single-nucleotide polymorphism (SNP) haplotypes among The Environmental Determinants of Diabetes in the Young (TEDDY) samples.

| | Tri-SNP Genotype | | | | | | | | | | | |
|---|---|---|---|---|---|---|---|---|---|---|---|---|
| | 010/010 | 010/101 | 101/101 | 000/010 | 001/010 | 000/101 | 000/000 | 000/001 | 001/101 | 011/101 | 010/011 | All |
| HLA | | | | | | | | | | | | |
| DR3/DR3 | 62 | 348 | 1177 | 13 | 2 | 11 | 0 | 0 | 2 | 1 | 0 | 1616 (20.8%) |
| DR3/DR4 | 408 | 2550 | 3 | 19 | 8 | 30 | 0 | 1 | 1 | 1 | 1 | 3022 (38.9%) |
| DR4/DR4 | 1480 | 5 | 3 | 39 | 3 | 1 | 0 | 0 | 0 | 0 | 0 | 1531 (19.7%) |
| DR4/DR8 | 14 | 0 | 0 | 1297 | 5 | 0 | 17 | 1 | 0 | 0 | 0 | 1334 (17.2%) |
| DR1/DR4 | 0 | 0 | 0 | 15 | 143 | 0 | 0 | 2 | 0 | 0 | 0 | 160 (2.1%) |
| DR4/DR13 | 0 | 0 | 0 | 0 | 56 | 0 | 0 | 1 | 0 | 0 | 0 | 57 (0.7%) |
| DR3/DR9 | 2 | 16 | 0 | 0 | 0 | 0 | 0 | 0 | 0 | 0 | 0 | 18 (0.2%) |
| DR4/DR9 | 13 | 1 | 0 | 0 | 0 | 0 | 0 | 0 | 0 | 0 | 0 | 14 (0.2%) |
| DR4/DR4*030 X/020 X | 4 | 0 | 0 | 0 | 0 | 0 | 0 | 0 | 0 | 0 | 0 | 4 (0.1%) |
| DR4/DR4*030 X/0304 | 3 | 0 | 0 | 0 | 0 | 0 | 0 | 0 | 0 | 0 | 0 | 3 (0.0%) |
| All | 1986 (25.6%) | 2920 (37.6%) | 1183 (15.2%) | 1383 (17.8%) | 217 (2.8%) | 42 (0.5%) | 17 (0.2%) | 5 (0.1%) | 3 (0.0%) | 2 (0.0%) | 1 (0.0%) | 7759 (100%) |

Swedish Better Diabetes Diagnosis (BDD) study, and the Swedish Diabetes Prediction in Skåne (DiPiS) study (*Aydemir et al., 2019*). The relative T1D risk of the 010/010 genotype, compared to the homozygous 101/101 genotype, was substantial in both the T1DGC and Swedish DR3/3 cohorts (odds ratio 4.65, *p*=1.69 × 10⁻¹³). This association was later independently replicated in a separate cohort of DR3-DQ2/DR3-DQ2 individuals from Finland (*Nygård et al., 2021*). These investigators reported that the 010/010 (AGG/AGG) or 010 /x (AGG/x) tri-SNP haplotypes were significantly more common in patients with T1D (OR = 1.70, CI 95%=1.15–2.51, *p*=0.007) than in non-diabetic controls.

Here, we report an expanded analysis of the effect of the tri-SNP on the appearance of islet autoimmunity and the subsequent progression to T1D using data from the large population enrolled in TEDDY study (*TEDDY Study Group, 2008*). In addition, because CD is also documented in the TEDDY study, the novel and unexpected discovery was made that the tri-SNP 101 genotype that confers T1D protection was associated with increased susceptibility to CD.

## Results

### Study population demographics

The TEDDY database analyzed here was frozen as of 31 October 2021 and the present analysis represents a median follow-up of 12.7 years (IQR = 10.9–14.4). 382 (5.0%) children had been diagnosed with T1D and 617 (8.0%) with CD. The TEDDY study is ongoing and about 75% of the research subjects have still to age out (15 years of age) of the study.

There were 7759 study subjects with HLA and tri-SNP typing data available. Ancestry distribution was 89.6% European (EUR, N=6953), 9.1% Ad Mixed American (AMR, N=707), 1% African (AFR, N=82), 0.2% South Asian (SAS, N=14), and <0.1% East Asian (EAS, N=3).

The distribution of tri-SNP haplotypes and HLA genotypes is shown in *Table 1*. As anticipated, given the study inclusion criteria, most (N=7503) participants were homozygous for either DR3 or DR4, heterozygous for DR3/DR4, or heterozygous for DR4/DR8. With respect to the distribution of tri-SNP haplotypes in this population, nearly all subjects had at least one 101 or 010 tri-SNP; only 22 (0.3%) did not. The 101 tri-SNP haplotype was observed almost exclusively coupled to the DR3-DQ2 haplotype; 99.6% (2357/2366) of the chromosomes from 101/101 homozygous individuals carried the DR3-DQ2 haplotype. Similarly, 101 is highly enriched in the DR3 population; 84% (2716/3232) of the chromosomes from DR3 homozygous individuals had 101, in line with our previous report (*Aydemir et al., 2019*). Approximately 99% of the DR3 homozygous population carried either 010 or 101 (3203/3232). We also observed that 98.2% of the DR4/DR4 homozygous population carried 010 (3007/3062).

Cox PH analysis requires each level of the categorical variables to have at least one event and one non-event data point. We removed the samples that belonged to categories with too few individuals (<20) to accommodate this requirement. These included samples from: HLA classes DR3/DR9 (n=18), DR4/DR9 (n=14), DR4/DR4*030 X/020 X (n=4), DR4/DR4*030 X/0304 (n=3); populations SAS (n=14) and EAS (n=4). 7703 samples remained in the final set. Descriptive characteristics of the final data set with respect to each outcome studied are provided in *Tables 2–7*.

### Tri-SNP association with T1D diagnosis

5.2% (397/7703) of the children were diagnosed with T1D. 65 were excluded from the time-to-T1D analysis due to missing data for controlled GWAS associations, leaving the final data set for this analysis at 7638, of which 5.2% (395) were diagnosed with T1D. Our model indicated that tri-SNP 101 reduced the risk of developing T1D (HR = 0.54, CI = 0.41–0.72, *p*=1.52e-5, *Figure 1A*). Known risk factors HLA DR3/DR4 genotype (HR = 3.20, CI = 2.18–4.70, *p*=3.16e-9) and having an affected FDR (HR = 3.11, CI = 2.42–3.98, *p*=3.75e-19) were the largest risk factors, as expected (*Krischer et al., 2017*; *Sharma et al., 2018*; *Figure 1A*, *Figure 1—figure supplement 1*, *Supplementary file 1A*). All six previously identified GWAS associations with T1D were also significantly associated with T1D diagnosis (*Krischer et al., 2017*; *Sharma et al., 2018*). While genetic ancestry and sex were not detected as significant risk factors, subjects from Finland were marginally at higher risk (HR = 1.34, *p*=0.04, *Figure 1—figure supplement 1*; *Table 8*).

Given the strong linkage between tri-SNP 101 and HLA DR3 haplotypes (*Table 1*), we performed a separate Cox PH regression using only DR3 homozygous samples for this and each subsequent study

**Table 2.** Descriptive characteristics of children with respect to the islet autoimmunity (IA) outcome in number (percentage).

| | IA | | |
|---|---|---|---|
| | No | Yes | All |
| **Sex** | | | |
| Female | 3388 (49.5) | 390 (45.5) | 3778 (49.0) |
| Male | 3457 (50.5) | 468 (54.5) | 3925 (51.0) |
| **POP** | | | |
| EUR | 6122 (89.4) | 796 (92.8) | 6918 (89.8) |
| AMR | 647 (9.5) | 56 (6.5) | 703 (9.1) |
| AFR | 76 (1.1) | 6 (0.7) | 82 (1.1) |
| **HLA Type** | | | |
| DR1/DR4 | 142 (2.1) | 18 (2.1) | 160 (2.1) |
| DR3/DR3 | 1486 (21.7) | 121 (14.1) | 1607 (20.9) |
| DR3/DR4 | 2599 (38.0) | 418 (48.7) | 3017 (39.2) |
| DR4/DR13 | 47 (0.7) | 10 (1.2) | 57 (0.7) |
| DR4/DR4 | 1371 (20.0) | 158 (18.4) | 1529 (19.8) |
| DR4/DR8 | 1200 (17.5) | 133 (15.5) | 1333 (17.3) |
| **Country** | | | |
| US | 2885 (42.1) | 297 (34.6) | 3182 (41.3) |
| SWE | 2019 (29.5) | 290 (33.8) | 2309 (30.0) |
| FIN | 1482 (21.7) | 214 (24.9) | 1696 (22.0) |
| GER | 459 (6.7) | 57 (6.6) | 516 (6.7) |
| **FDR** | | | |
| 0 | 6155 (89.9) | 691 (80.5) | 6846 (88.9) |
| 1 | 690 (10.1) | 167 (19.5) | 857 (11.1) |
| **All** | 6845 (88.9) | 858 (11.1) | 7703 (100) |

outcome. When the analysis was restricted to DR3-DQ2 homozygous children, the protective effect of the 101 tri-SNP was even stronger (HR = 0.35, CI = 0.22–0.56, p=1.06e-5, *Figure 1A*), confirming our initial report (*Aydemir et al., 2019*) and the confirmatory Finnish report (*Nygård et al., 2021*). Among DR3 homozygous individuals, FDR (HR = 2.86, CI = 1.22–6.67, p=0.02) and rs2476601 (PTPN22, HR = 2.51, CI = 1.38–4.56, p=0.003) remained significant risk factors (*Figure 1—figure supplement 2*, *Supplementary file 1B*).

## Islet autoantibody (IA) development

At least one IA (the primary TEDDY outcome variable) was detected in 858/7703 (11.1%) children from the dataset. 89 children had missing data for controlled GWAS genotypes and were not included in the Cox PH model for IA outcome. Our model showed a significantly reduced independent risk associated with the tri-SNP 101 haplotype (HR = 0.71, CI = 0.58–0.87, p=0.001), each protective allele conferring 29% reduction in risk of developing islet autoantibodies (*Figure 1B*). Among the controlled covariates, having a first-degree relative (FDR) with T1D (HR = 2.16, CI = 1.78–2.62, p=5.76e-15, *Figure 1B*), HLA DR3/DR4 genotype (HR = 1.83, CI = 1.41–2.39, p=6.96e-6, *Figure 1—figure supplement 3*, *Supplementary file 1C*) were significant risk factors for IA; and male subjects were found to be at marginally higher risk of developing IA (HR = 1.19, CI = 1.03–1.36, p=0.01), consistent with previous TEDDY reports (*Krischer et al., 2017*; *Sharma et al., 2018*).

**Table 3.** Descriptive characteristics of children with respect to type 1 diabetes diagnosis (T1D) outcome in number (percentage).

| | T1D | | |
| --- | --- | --- | --- |
| Sex | No | Yes | All |
| Female | 3597 (49.2) | 181 (45.6) | 3778 (49.0) |
| Male | 3709 (50.8) | 216 (54.4) | 3925 (51.0) |
| POP | | | |
| EUR | 6552 (89.7) | 366 (92.2) | 6918 (89.8) |
| AMR | 674 (9.2) | 29 (7.3) | 703 (9.1) |
| AFR | 80 (1.1) | 2 (0.5) | 82 (1.1) |
| HLA Type | | | |
| DR1/DR4 | 146 (2.0) | 14 (3.5) | 160 (2.1) |
| DR3/DR3 | 1571 (21.5) | 36 (9.1) | 1607 (20.9) |
| DR3/DR4 | 2797 (38.3) | 220 (55.4) | 3017 (39.2) |
| DR4/DR13 | 51 (0.7) | 6 (1.5) | 57 (0.7) |
| DR4/DR4 | 1457 (19.9) | 72 (18.1) | 1529 (19.8) |
| DR4/DR8 | 1284 (17.6) | 49 (12.3) | 1333 (17.3) |
| Country | | | |
| US | 3034 (41.5) | 148 (37.3) | 3182 (41.3) |
| SWE | 2204 (30.2) | 105 (26.4) | 2309 (30.0) |
| FIN | 1591 (21.8) | 105 (26.4) | 1696 (22.0) |
| GER | 477 (6.5) | 39 (9.8) | 516 (6.7) |
| FDR | | | |
| 0 | 6560 (89.8) | 286 (72.0) | 6846 (88.9) |
| 1 | 746 (10.2) | 111 (28.0) | 857 (11.1) |
| All | 7306 (94.8) | 397 (5.2) | 7703 (100) |

Subjects from Finland (HR = 1.34, CI = 1.11–1.63, $p$=0.002) and Sweden (HR = 1.25, CI = 1.06–1.48, $p$=0.008) were at slightly higher risk than subjects from the United States (*Figure 1—figure supplement 3*). Seven GWAS loci reported to be associated with IA in the TEDDY cohort (*Krischer et al., 2017*; *Sharma et al., 2018*) were confirmed to be significantly associated and in the same direction (protective vs risk) with IA development in our analysis.

When analysis was restricted to DR3 homozygous individuals, the protective effect of the tri-SNP 101 haplotype against development of IA was stronger (HR = 0.58, CI = 0.42–0.80, $p$=0.001), each allele conferring 42% reduction in risk. Having an FDR with T1D remained a significant additional risk factor within this group (HR = 2.42, CI = 1.49–3.93, $p$=3.37e-4, *Figure 1B*), but other controlled covariates were no longer significant (*Figure 1—figure supplement 4*, *Supplementary file 1D*).

### First appearing islet autoantibody

We assessed the effect of tri-SNP 101 on two T1D autoimmunity endotypes, based on the first appearing islet autoantibody being IAA or glutamic acid decarboxylase autoantibody (GADA) in separate models, controlling for the same covariates as in the IA analysis. Although tri-SNP-101 trended towards protection against GADA-first outcome, this effect did not reach statistical significance either in the entire dataset (HR = 0.87, CI = 0.64–1.17, $p$=0.34, *Figure 1D*, *Figure 1—figure supplement 5*, *Supplementary file 1E*) or in DR3-DQ2 homozygotes (HR = 0.71, CI = 0.47–1.07, $p$=0.10, *Figure 1D*, *Figure 1—figure supplement 6*, *Supplementary file 1F*). IAA-first risk was significantly reduced by tri-SNP 101 both in the entire data set (HR = 0.64, CI = 0.45–0.91, $p$=0.01, *Figure 1C*, *Figure 1—figure*

**Table 4.** Descriptive characteristics of children with respect to glutamic acid decarboxylase autoantibody (GADA)-first appearing antibody outcome in number (percentage).

| | GADA first | | |
| --- | --- | --- | --- |
| | No | Yes | All |
| **Sex** | | | |
| Female | 3601 (49.2) | 177 (46.3) | 3778 (49.0) |
| Male | 3720 (50.8) | 205 (53.7) | 3925 (51.0) |
| **POP** | | | |
| EUR | 6565 (89.7) | 353 (92.4) | 6918 (89.8) |
| AMR | 678 (9.3) | 25 (6.5) | 703 (9.1) |
| AFR | 78 (1.1) | 4 (1.0) | 82 (1.1) |
| **HLA Type** | | | |
| DR1/DR4 | 157 (2.1) | 3 (0.8) | 160 (2.1) |
| DR3/DR3 | 1522 (20.8) | 85 (22.3) | 1607 (20.9) |
| DR3/DR4 | 2828 (38.6) | 189 (49.5) | 3017 (39.2) |
| DR4/DR13 | 55 (0.8) | 2 (0.5) | 57 (0.7) |
| DR4/DR4 | 1472 (20.1) | 57 (14.9) | 1529 (19.8) |
| DR4/DR8 | 1287 (17.6) | 46 (12.0) | 1333 (17.3) |
| **Country** | | | |
| US | 3036 (41.5) | 146 (38.2) | 3182 (41.3) |
| SWE | 2164 (29.6) | 145 (38.0) | 2309 (30.0) |
| FIN | 1622 (22.2) | 74 (19.4) | 1696 (22.0) |
| GER | 499 (6.8) | 17 (4.5) | 516 (6.7) |
| **FDR** | | | |
| No | 6531 (89.2) | 315 (82.5) | 6846 (88.9) |
| Yes | 790 (10.8) | 67 (17.5) | 857 (11.1) |
| **All** | 7321 (95.0) | 382 (5.0) | 7703 (100) |

supplement 7, *Supplementary file 1G*) and the DR3-DQ2 homozygotes (HR = 0.48, CI = 0.25–0.91, $p$=0.03, *Figure 1C*, *Figure 1—figure supplement 8*, *Supplementary file 1H*). We note that since the incidence of IAA-first among DR3-DQ2 homozygotes at 1.8% (29/1578) is relatively low, samples from an African genetic background (N=32, incidence = 0), and one low-frequency GWAS locus were excluded from this analysis to accommodate the model requirements.

## Celiac Diagnosis (CD)

Among the study subjects, 617 (8.0%) were diagnosed with CD. 24 children had both T1D and CD which corresponded to 3.9% of all CD samples and 6.0% of all T1D samples. 6530 children had complete information on all covariates for CD and these were used in the Cox PH time-to-CD analysis.

In contrast to its protective effect against T1D, tri-SNP 101 was a significant risk factor (HR = 1.32, CI = 1.06–1.64, $p$=0.01, *Figure 1E*) for CD. As previously reported, female sex, FDR, gluten intake, and HLA-DR3, and HLA-DR4 haplotypes were significant risk factors (*Andrén Aronsson et al., 2019*), (*Figure 1E*, *Figure 1—figure supplement 9*, *Supplementary file 1I*). Previously identified GWAS associations (*Sharma et al., 2016*) were also observed in our analysis. Individuals assigned to AMR genetic ancestry were at lower risk of CD (HR = 0.62, $p$=0.04) compared to the baseline EUR population (*Figure 1—figure supplement 9*, *Supplementary file 1I*).

**Table 5.** Descriptive characteristics of children with respect to insulin autoantibody (IAA)-first appearing antibody outcome in number (percentage).

| | IAA first | | |
| --- | --- | --- | --- |
| | No | Yes | All |
| **Sex** | | | |
| Female | 3636 (49.2) | 142 (45.4) | 3778 (49.0) |
| Male | 3754 (50.8) | 171 (54.6) | 3925 (51.0) |
| **POP** | | | |
| EUR | 6624 (89.6) | 294 (93.9) | 6918 (89.8) |
| AMR | 685 (9.3) | 18 (5.8) | 703 (9.1) |
| AFR | 81 (1.1) | 1 (0.3) | 82 (1.1) |
| **HLA Type** | | | |
| DR1/DR4 | 151 (2.0) | 9 (2.9) | 160 (2.1) |
| DR3/DR3 | 1578 (21.4) | 29 (9.3) | 1607 (20.9) |
| DR3/DR4 | 2870 (38.8) | 147 (47.0) | 3017 (39.2) |
| DR4/DR13 | 52 (0.7) | 5 (1.6) | 57 (0.7) |
| DR4/DR4 | 1471 (19.9) | 58 (18.5) | 1529 (19.8) |
| DR4/DR8 | 1268 (17.2) | 65 (20.8) | 1333 (17.3) |
| **Country** | | | |
| US | 3082 (41.7) | 100 (31.9) | 3182 (41.3) |
| SWE | 2215 (30.0) | 94 (30.0) | 2309 (30.0) |
| FIN | 1597 (21.6) | 99 (31.6) | 1696 (22.0) |
| GER | 496 (6.7) | 20 (6.4) | 516 (6.7) |
| **FDR** | | | |
| No | 6600 (89.3) | 246 (78.6) | 6846 (88.9) |
| Yes | 790 (10.7) | 67 (21.4) | 857 (11.1) |
| | | | |
| **All** | 7390 (95.9) | 313 (4.1) | 7703 (100) |

The increased CD risk associated with the 101 tri-SNP haplotype remained significant (HR = 1.32, CI = 1.02–1.70, p=0.04) when the analysis was restricted to HLA-DR3 homozygotes (*Figure 1E*, *Figure 1—figure supplement 10*, *Supplementary file 1J*).

## Celiac Disease Autoimmunity (CDA) Development

6709 children had CDA data available. 19.2% (1294) were positive for CDA. 174 showed both CDA and IA corresponding to 13.4% of all CDA and 21.3% of all IA individuals. 6557 children had complete information for all covariates used in the statistical model.

Similar to its effect on CD and contrary to T1D and IA, the tri-SNP 101 haplotype was a significant risk factor (HR = 1.23, CI = 1.06–1.44, p=0.008) for CDA (*Figure 1F*). Other risk factors included FDR with celiac disease, female sex, HLA-DR3 and DR4 haplotypes, gluten intake, and European genetic ancestry, all of which have been reported previously (*Andrén Aronsson et al., 2019*; *Singh et al., 2015*; *Figure 1F*, *Figure 1—figure supplement 11*, *Supplementary file 1K*).

Tri-SNP 101 remained a risk factor when the analysis was restricted to DR3-DQ2 homozygotes (HR = 1.23, CI = 1.02–1.48, p=0.03, *Figure 1—figure supplement 12*, *Supplementary file 1L*).

**Table 6.** Descriptive characteristics of children with respect to celiac disease diagnosis (CD) outcome in number (percentage).

| | CD | | |
|---|---|---|---|
| | No | Yes | All |
| **Sex** | | | |
| Female | 3413 (48.2) | 365 (59.2) | 3778 (49.0) |
| Male | 3673 (51.8) | 252 (40.8) | 3925 (51.0) |
| **POP** | | | |
| EUR | 6325 (89.3) | 593 (96.1) | 6918 (89.8) |
| AMR | 681 (9.6) | 22 (3.6) | 703 (9.1) |
| AFR | 80 (1.1) | 2 (0.3) | 82 (1.1) |
| **HLA Type** | | | |
| DR1/DR4 | 155 (2.2) | 5 (0.8) | 160 (2.1) |
| DR3/DR3 | 1308 (18.5) | 299 (48.5) | 1607 (20.9) |
| DR3/DR4 | 2805 (39.6) | 212 (34.4) | 3017 (39.2) |
| DR4/DR13 | 54 (0.8) | 3 (0.5) | 57 (0.7) |
| DR4/DR4 | 1444 (20.4) | 85 (13.8) | 1529 (19.8) |
| DR4/DR8 | 1320 (18.6) | 13 (2.1) | 1333 (17.3) |
| **Country** | | | |
| US | 2955 (41.7) | 227 (36.8) | 3182 (41.3) |
| SWE | 2049 (28.9) | 260 (42.1) | 2309 (30.0) |
| FIN | 1594 (22.5) | 102 (16.5) | 1696 (22.0) |
| GER | 488 (6.9) | 28 (4.5) | 516 (6.7) |
| **FDR** | | | |
| No | 6447 (96.4) | 496 (80.7) | 6943 (95.1) |
| Yes | 242 (3.6) | 119 (19.3) | 361 (4.9) |
| **All** | 7086 (92.0) | 617 (8.0) | 7703 (100) |

## Comparison to HLA-DR3 B8 extended haplotype

It has been reported that two conserved extended haplotypes carrying DR3 (B8-DR3 and B18-DR3) differentially contribute to the risk for T1D and CD in the Basque population (*Bilbao et al., 2006*). Our analysis shows that the tri-SNP 101 haplotype is weakly correlated with the CD-predisposing B8-DR3 extended haplotype ($R^2$=0.32 for DR3 homozygotes, data not shown). However, when the Cox proportional hazard model was fitted using the B8 dosage instead of the tri-SNP, the associations with T1D (B8 HR = 0.64 vs tri-SNP HR = 0.54, *Supplementary file 1M*) and IA (B8 HR = 0.80 vs tri-SNP HR = 0.71, *Supplementary file 1N*) were weaker, and no significant association was detected either for CD (HR = 1.11, *p*=0.12, *Supplementary file 1O*) or for CDA (HR = 1.07, *p*=0.20, *Supplementary file 1P*).

## Differential Gene Expression

We explored the leukocyte gene expression differences within 538 samples from 129 DR3-DQ2 homozygous individuals of our study cohort. Expression of 68 genes were found to be significantly affected by the tri-SNP 101 haplotype (adjusted-p <0.01, *Supplementary file 1Q*). Eight of 10 of the most significant changes were affected genes located in the MHC loci.

**Table 7.** Descriptive characteristics of children with respect to celiac disease autoimmunity (CDA) outcome in number (percentage).

| | CDA | | |
| --- | --- | --- | --- |
| | No | Yes | All |
| **Sex** | | | |
| Female | 2555 (47.2) | 735 (56.8) | 3290 (49.0) |
| Male | 2860 (52.8) | 559 (43.2) | 3419 (51.0) |
| **POP** | | | |
| EUR | 4855 (89.7) | 1237 (95.6) | 6092 (90.8) |
| AMR | 515 (9.5) | 55 (4.3) | 570 (8.5) |
| AFR | 45 (0.8) | 2 (0.2) | 47 (0.7) |
| **HLA Type** | | | |
| DR1/DR4 | 134 (2.5) | 8 (0.6) | 142 (2.1) |
| DR3/DR3 | 870 (16.1) | 530 (41.0) | 1400 (20.9) |
| DR3/DR4 | 2146 (39.6) | 503 (38.9) | 2649 (39.5) |
| DR4/DR13 | 46 (0.8) | 4 (0.3) | 50 (0.7) |
| DR4/DR4 | 1130 (20.9) | 193 (14.9) | 1323 (19.7) |
| DR4/DR8 | 1089 (20.1) | 56 (4.3) | 1145 (17.1) |
| **Country** | | | |
| US | 2232 (41.2) | 471 (36.4) | 2703 (40.3) |
| SWE | 1588 (29.3) | 479 (37.0) | 2067 (30.8) |
| FIN | 1262 (23.3) | 272 (21.0) | 1534 (22.9) |
| GER | 333 (6.1) | 72 (5.6) | 405 (6.0) |
| **FDR** | | | |
| No | 5118 (96.5) | 1119 (86.9) | 6237 (94.6) |
| Yes | 186 (3.5) | 169 (13.1) | 355 (5.4) |
| **All** | 5415 (80.7) | 1294 (19.3) | 6709 (100) |

The expression of complement system genes C4A and C4B was inversely affected by the tri-SNP 101 dosage (*Figure 2*). Whereas C4A gene expression decreased (*p*=1.46E-22), C4B significantly increased (*p*=5.32E-13) with increasing tri-SNP 101 dosage.

## Copy Number Variation in C4 Locus

The C4 locus has been reported to have frequent copy number variation and associations with autoimmunity (*Li et al., 2017*). We analyzed the whole genome sequencing data of 188 DR3-DQ2 homozygous individuals from our cohort to investigate whether there are copy number changes in C4A or C4B. Read coverage data showed reduced coverage of C4 genes in some samples indicating the presence of gene deletions (*Figure 3—figure supplement 1A*). Due to extensive sequence identity between C4A and C4B, determination of C4A and C4B copy numbers was based only on reads mapping uniquely to either gene (*Figure 3*). We identified frequent C4A deletions: 56.9% (107/188) of the samples were C4A null, and 23.4% had only a single C4A (*Table 9*). All 107 C4A null samples were homozygous for tri-SNP 101 genotype indicating a strong association (chi-squared *p*-value = 5.59E-38). C4B deletions were also common, although not as frequent as C4A. There was only one C4B null sample and 22.8% (43/188) of the samples had a single C4B gene (*Table 10*). We observed a positive association between tri-SNP 101 and C4B copy number (chi-squared *p*-value = 1.89E-20)

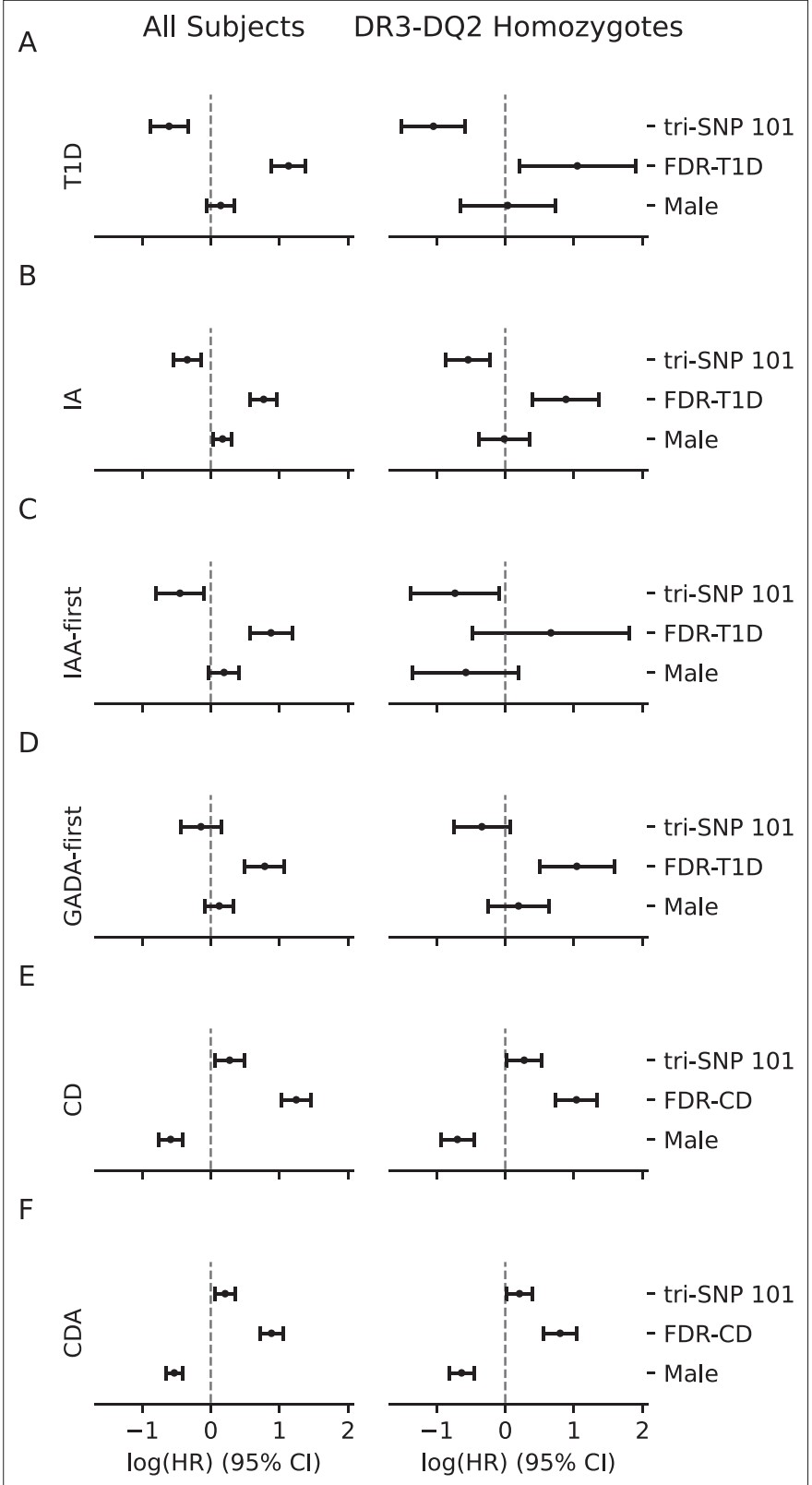

**Figure 1.** Cox Proportional Hazard (PH) regression results for all tested outcomes. Log hazard ratios (log(HR)) and 95% confidence intervals of the tri-single-nucleotide polymorphism (SNP) 101 haplotype as well as the known risk factors first-degree relative (FDR) and sex from the Cox PH models for outcomes type 1 diabetes (T1D) (**A**), islet antigen (IA) (**B**), insulin autoantibody (IAA)-first (**C**), glutamic acid decarboxylase autoantibody (GADA)-first

*Figure 1 continued on next page*

*Figure 1 continued*

(**D**), celiac disease (CD) (**E**), and celiac disease autoimmunity (CDA) (**F**), using the entire cohort (left) or only the DR3-DQ2 homozygote individuals (right). Dashed vertical line at 0 indicating an HR of 1 (log(HR)=0), i.e., no effect on risk. Left side of the vertical line indicates reduced risk vs increased risk on the right side. Whiskers indicate 95% CI around HR. The model assesses the independent risk/protection afforded by each covariate compared to the baseline for categorical covariates FDR and sex for which the baselines are having no FDR and female sex, respectively. Tri-SNP 101 is modeled numerically, so the HR reported is per each additional 101 allele.

The online version of this article includes the following figure supplement(s) for figure 1:

**Figure supplement 1.** Complete model output for type 1 diabetes (T1D) outcome.

**Figure supplement 2.** Complete model output for type 1 diabetes (T1D) outcome using only DR3-DQ2 homozygotes.

**Figure supplement 3.** Complete model output for islet antigen (IA) outcome using all samples.

**Figure supplement 4.** Complete model output for islet antigen (IA) outcome using only DR3-DQ2 homozygotes.

**Figure supplement 5.** Complete model output for glutamic acid decarboxylase autoantibody (GADA)-first outcome.

**Figure supplement 6.** Complete model output for glutamic acid decarboxylase autoantibody (GADA)-first outcome using only DR3-DQ2 homozygotes.

**Figure supplement 7.** Complete model output for insulin autoantibody (IAA)-first outcome.

**Figure supplement 8.** Complete model output for insulin autoantibody (IAA)-first outcome using only DR3-DQ2 homozygotes.

**Figure supplement 9.** Complete model output for celiac disease (CD) outcome.

**Figure supplement 10.** Complete model output for celiac disease (CD) outcome using only DR3-DQ2 homozygotes.

**Figure supplement 11.** Complete model output for celiac disease autoimmunity (CDA) outcome.

**Figure supplement 12.** Complete model output for celiac disease autoimmunity (CDA) outcome using only DR3-DQ2 homozygotes.

in contrast to the negative correlation with C4A. We also noted that 89.4% (168/188) of samples had total C4 gene copy number <4.

## Discussion

We report confirmation of the tri-SNP association with T1D in a much larger, independent cohort. In addition, we show that the 101 tri-SNP haplotype is also associated with reduced risk of developing IA. Although the primary outcome of the TEDDY study was IA, it has been recognized in multiple studies that there may be two distinct 'endotypes' of T1D autoimmunity based on the timing of emergence of specific autoantibodies (*Battaglia et al., 2020*; *Johnson et al., 2021*; *Krischer et al., 2015*). One of the proposed endotypes is characterized by the appearance of IAA early on as the first marker of IA, termed 'IAA-first', and it is associated with HLA DR4-DQ8 haplotype. The second endotype is marked by the emergence at a later time of GAD autoantibodies as the first IA marker, termed 'GADA-first,' is associated with DR3-DQ2 haplotype (*Krischer et al., 2015*). We show that the tri-SNP is significantly associated with protection from the IAA-first outcome but not the GADA-first outcome, indicating that the main T1D protective effect of tri-SNP may be due to delay or prevention of insulin autoantibody generation. This may explain the low incidence of the IAA-first endotype among DR3-DQ2 carrying individuals, given the enrichment of the protective tri-SNP allele on the DR3-DQ2 chromosomes.

Unexpectedly, we discovered that the T1D-protective tri-SNP 101 haplotype is associated with *increased* risk for both autoantibody development (CDA) and clinical CD. This finding is surprising because CD and T1D are thought to share genetic risk factors not only because of the increased co-occurrence of the two pathologies but also because DR3-DQ2 and DR4-DQ8 are major genetic loci for susceptibility for both disorders. An extended HLA-DR3 haplotype (B8) was described in the literature as having a differential impact on T1D and CD development in a small sample set of the Basque population (*Bilbao et al., 2006*) but this was not replicated in our sample set. Therefore, the opposite

**Table 8.** Previously published Genome-wide association studies (GWAS) associations.

Single-nucleotide polymorphisms (SNPs) used as covariates in our CoxPH analysis. Celiac disease (CD), celiac disease autoimmunity (CDA), type 1 diabetes (T1D), and islet antigen (IA) columns indicating whether the SNP has been shown to be associated with that outcome and hence used in the model (yes) or not (no). Statistically significant hazard ratios for the associated outcome are also provided under HR columns, protective associations in bold.

| SNP | Locus | CD | CDA | T1D | IA | Publication | HR CD | HR CDA | HR T1D | HR IA |
|---|---|---|---|---|---|---|---|---|---|---|
| rs4851575 | IL18R1, IL18RAP | yes | no | no | no | *Sharma et al., 2016* | 1.45 | | | |
| rs114569351 | PLEK, FBXO48 | yes | no | no | no | *Sharma et al., 2016* | 2.64 | | | |
| rs12493471 | CCR9, LZTFL1, CXCR6 | yes | no | no | no | *Sharma et al., 2016* | 1.40 | | | |
| rs1054091 | RSPH3, TAGAP | yes | no | no | no | *Sharma et al., 2016* | 1.59 | | | |
| rs72704176 | ASH1L | yes | no | no | no | *Sharma et al., 2016* | 2.26 | | | |
| rs3771689 | BAZ2B | yes | no | no | no | *Sharma et al., 2016* | **0.56** | | | |
| rs13014907 | ZNF804A | yes | no | no | no | *Sharma et al., 2016* | 2.46 | | | |
| rs11739460 | TCOF1 | yes | no | no | no | *Sharma et al., 2016* | 1.41 | | | |
| rs77532435 | GRB10 | yes | no | no | no | *Sharma et al., 2016* | 2.05 | | | |
| rs6967298 | AUTS2 | yes | no | no | no | *Sharma et al., 2016* | **0.61** | | | |
| rs61751041 | LAMB1 | yes | no | no | no | *Sharma et al., 2016* | 2.23 | | | |
| rs2409747 | XKR6 | yes | yes | no | no | *Sharma et al., 2016* | 1.58 | 1.37 | | |
| rs12990970 | NPM1P33, CTLA4 | no | yes | no | no | *Sharma et al., 2016* | | **0.76** | | |
| rs11709472 | LPP | no | yes | no | no | *Sharma et al., 2016* | | **0.80** | | |
| rs72717025 | FCGR2A | no | yes | no | no | *Sharma et al., 2016* | | 1.84 | | |
| rs114157400 | BANK1 | no | yes | no | no | *Sharma et al., 2016* | | 1.62 | | |
| rs117561283 | IFNG | no | yes | no | no | *Sharma et al., 2016* | | 1.81 | | |
| rs8013918 | FOS | no | yes | no | no | *Sharma et al., 2016* | | **0.80** | | |
| rs73043122 | RNASET2, MIR3939 | no | no | yes | no | *Sharma et al., 2018* | | | 3.35 | |
| rs113306148 | PLEKHA1, MIR3941 | no | no | yes | no | *Sharma et al., 2018* | | | 3.06 | |
| rs428595 | PPIL2 | no | no | yes | yes | *Sharma et al., 2018* | | | 3.42 | 2.46 |
| rs1004446 | INS | no | no | yes | yes | *Krischer et al., 2017* | | | **0.55** | **0.67** |
| rs2476601 | PTPN22 | no | no | yes | yes | *Krischer et al., 2017* | | | 1.91 | 1.73 |
| rs2292239 | ERBB3 | no | no | yes | yes | *Krischer et al., 2017* | | | 1.68 | 1.45 |
| rs3184504 | SH2B3 | no | no | no | yes | *Krischer et al., 2017* | | | | 1.40 |
| rs9934817 | RBFOX1 | no | no | no | yes | *Sharma et al., 2018* | | | | 2.66 |
| rs11705721 | PXK, PDHB | no | no | no | yes | *Sharma et al., 2018* | | | | 1.41 |

effects of the tri-SNP haplotype on the risk of T1D and CD in our cohort are novel and independent of the extended B8 haplotype.

Tri-SNP is a valuable addition to the collection of known genetic and environmental factors that are used to predict T1D and CD (*Bonifacio et al., 2018*; *Romanos et al., 2014*). Knowing whether a high-risk HLA child is more likely to develop T1D or CD may aid in decisions about early therapeutic or preventive interventions, such as a strict gluten-free diet or preventive anti-CD3 antibody treatment

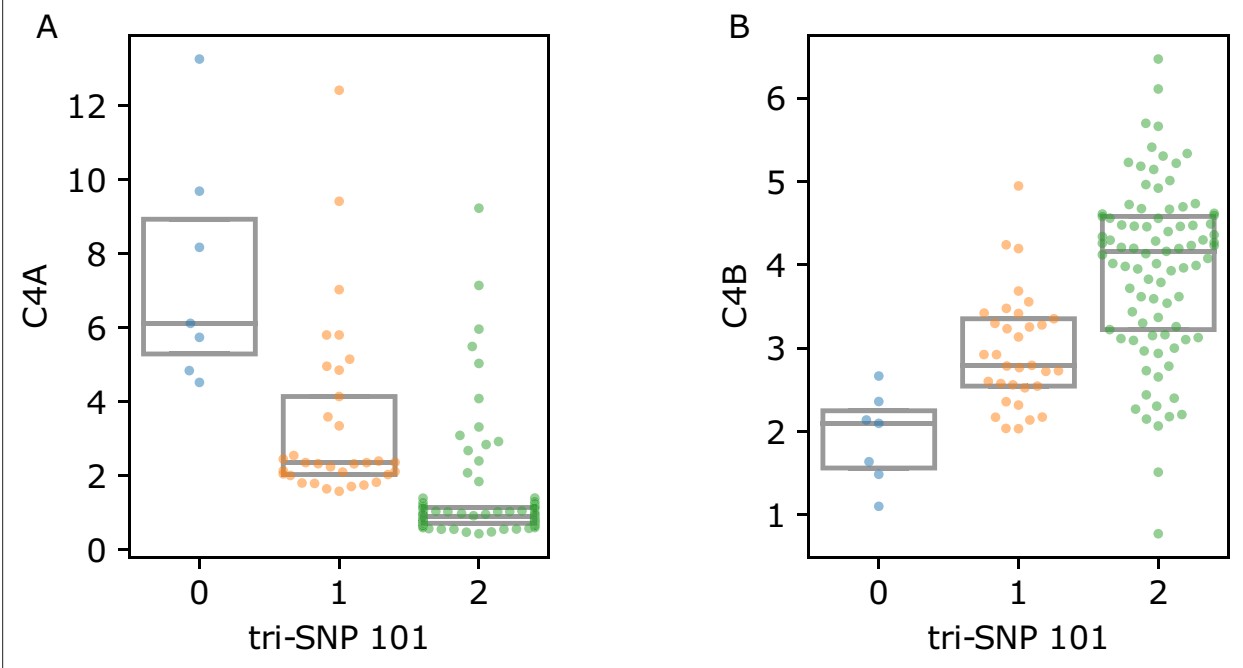

**Figure 2.** C4 gene expression values with respect to tri-single-nucleotide polymorphism (SNP). Count per million (CPM) values in 129 DR3 homozygous individuals showing decreasing C4A and increasing C4B gene expression as tri-SNP 101 allele count increases. Each point represents the median CPM value of multiple samples from one individual. Boxes represent the interquartile range (IQR) and midlines mark the median value.

(*Herold et al., 2019*). Our findings advance our understanding of complex polygenic diseases and should provide a more precise risk/benefit assessment of potential treatments based on data.

Furthermore, we investigated potential mechanisms underlying the differential association of tri-SNP with T1D and CD by analyzing gene expression differences based on the tri-SNP genotype. Complement system genes C4A and C4B emerged as intriguing candidates whose expression changed in opposite direction with respect to tri-SNP genotype. C4 locus has been reported to have frequent copy number variation (*Li et al., 2017*). Therefore, we explored whether the gene expression changes we observe may be due to changes in gene copy numbers (GCN). Indeed, tri-SNP 101, which is associated with decreased C4A and increased C4B gene expression, was also associated with frequent C4A gene deletions and increased C4B GCN. We also observed that close to 90% of our samples were missing at least one C4 gene. Reduced C4 GCN have been associated with the development of autoimmune diseases, such as systemic lupus erythematosus (*Pereira et al., 2016*), juvenile dermatomyositis (*Lintner et al., 2016*), and rheumatoid arthritis (*Rigby et al., 2012*). In addition, lower serum levels of C4 protein have been suggested to predispose to T1D (*Vergani et al., 1983*). Taken together, reduced total C4 GCN frequent in our high-risk HLA cohort is in line with the reports of increased susceptibility of low C4 to autoimmune diseases, and the different disease outcomes based on the tri-SNP haplotype may be, in part, due to the absence of specific C4 genes carried on that haplotype. Although the link between tri-SNP and C4 GCN is intriguing, independent association analyses for C4 genes and diseases were not possible due to the small number of samples with the GCN data. In addition, given the presence of other differentially expressed genes in the MHC locus, it is possible that the differential risk marked by tri-SNP is due to multiple factors.

## Methods
### Study population
Data were obtained from participants in the TEDDY prospective cohort study, whose primary goal is to identify environmental causes of T1D. It includes three clinical research centers in the United States (Colorado, Georgia/Florida, Washington) and three in Europe (Finland, Germany, and Sweden), and has been described in detail (*The TEDDY Study Group, 2007*; *TEDDY Study Group, 2008*). Briefly,

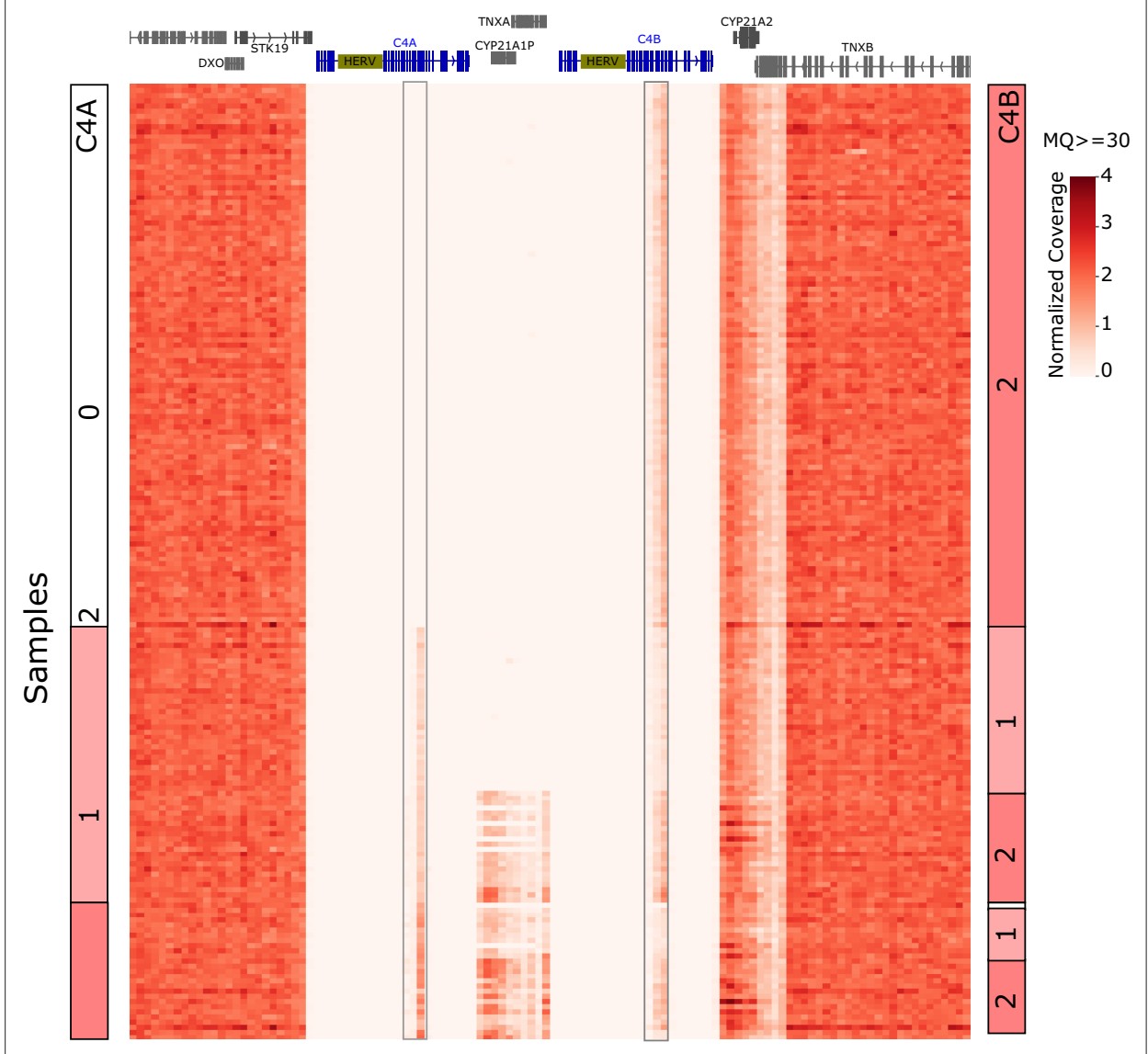

**Figure 3.** Unique sequence read coverage in C4 region and copy number calls. Uniquely mapping read coverage from whole genome sequencing (WGS) data of 188 homozygous DR3-DQ2 individuals. C4A and C4B genes share extensive sequence identity along the genes except a ~3 kilobase region indicated with boxes. Reads mapping to these regions were used to estimate C4A (left column) and C4B (right column) copy numbers per sample. Samples were sorted based on C4A copy numbers. A maximum value of 4 was used for the heatmap to moderate high outlier values.

The online version of this article includes the following figure supplement(s) for figure 3:

**Figure supplement 1.** Sequence read coverage in C4 region.

**Table 9.** Estimated C4A gene copy number with respect to tri-single-nucleotide polymorphism (SNP) haplotype.

| | tri-SNP 101 | | |
|---|---|---|---|
| C4A copy number | 0 | 1 | 2 |
| 0 | 0 | 0 | 107 |
| 1 | 0 | 44 | 10 |
| 2 | 9 | 9 | 7 |
| 3 | 1 | 0 | 1 |

**Table 10.** Estimated C4B gene copy number with respect to tri-single-nucleotide polymorphism (SNP) haplotype.

| | tri-SNP 101 | | |
|---|---|---|---|
| C4B copy number | 0 | 1 | 2 |
| 0 | 1 | 0 | 0 |
| 1 | 5 | 34 | 4 |
| 2 | 4 | 17 | 119 |
| 3 | 0 | 2 | 2 |

participation in the study was offered to the parents of all newborns at each research center. Inclusion criteria were based on HLA genotype and are described below. Separate written informed consents were obtained for all study participants from a parent or primary caretaker for genetic screening and for participation in prospective follow-up. The study was approved by local Institutional Review Boards and is monitored by an external advisory board formed by the National Institutes of Health.

## Inclusion criteria

Genotype screening (*Hagopian et al., 2011*) was conducted using either a dried blood spot punch or a small volume whole blood lysate specimen as described (*Dantonio et al., 2010*). Infants were eligible for the study if they had 2 high-risk HLA genotypes or had a first-degree relative with T1D and at least one high-risk HLA genotype. Details of these inclusion criteria are as follows:

Infants from the general population were eligible for the study if they had any one of the following HLA genotypes (excluding those with DR4 subtype DRB1*04:03):

1. DR3-DQA1*05:01-DQB1*02:01/DR4-DQA1*03:0X-DQB1*03:02
2. DR4-DQA1*03:0X-DQB1*03:02/DR4-DQA1*03:0X-DQB1*03:02
3. DR4-DQA1*03:0X-DQB1*03:02/DR8-DQA1*04:01-DQB1*04:02
4. DR3-DQA1*05:01-DQB1*02:01/DR3-DQA1*05:01-DQB1*02:01

Infants with a first-degree relative with T1D were eligible for enrollment if they had any of the following HLA genotypes:

1. DR4-DQA1*03:0X-DQB1*03:02/DR3-DQA1*05:01-DQB1*02:01
2. DR4-DQA1*03:0X-DQB1*03:02/DR4-DQA1*03:0X-DQB1*03:02
3. DR4-DQA1*03:0X-DQB1*03:02/DR8-DQA1*04:01-DQB1*04:02
4. DR3-DQA1*05:01-DQB1*02:01/DR3-DQA1*05:01-DQB1*02:01
5. DR4-DQA1*03:0X-DQB1*03:02/DR4-DQA1*03:0X-DQB1*02:0 X
6. DR4-DQA1*03:0X-DQB1*03:02/DR12-DQA1*01:01-DQB1*05:01
7. DR4-DQA1*03:0X-DQB1*03:02/DR13-DQA1*01:02-DQB1*06:04
8. DR4-DQA1*03:0X-DQB1*03:02/DR4-DQA1*03:0X-DQB1*03:04
9. DR4-DQA1*03:0X-DQB1*03:02/DR9-DQA1*03:0X-DQB1*03:03
10. DR3-DQA1*05:01-DQB1*02:01/DR9-DQA1*03:0X-DQB1*03:03

## HLA nomenclature

Explanation of HLA nomenclature used is as follows: DR3=DRB1*03:01; DR4=DRB1*04:01, *04:02, *04:04, *04:05, or *04:07; DQA1*03:0X=any DQA1*03; DQB1*03:02=DQB1*03:02 or DQB1*03:04; DQB1*02:0X=any DQB1*02; DR8=any DRB1*08; DR9=DRB1*09:01; DR12=any DRB1*12; DR13=any DRB1*13.

## HLA typing

Screening blood samples were generally obtained at birth from cord blood. Other potential participants, especially first-degree relatives of T1D participants at the Washington site, were screened using heel stick capillary samples up to the age of 4 months. This exception was made to maximize the number of newborn relatives participating in this study. After polymerase chain reaction (PCR) amplification of exon 2 of the HLA Class II gene (DRB1, DQA1, or DQB1), alleles are identified either by direct sequencing, oligonucleotide probe hybridization, or other genotyping techniques as

described (*Hagopian et al., 2011*). Additional typing to sufficiently identify certain DR-DQ haplotypes was specific to each clinical center.

When a TEDDY participant was 9–12 months of age, HLA status was confirmed by genotyping at increased resolution of HLA-DRB1, DQA1, and DQB1 at the central HLA reference laboratory at Roche Molecular Systems, Oakland, CA (*Erlich et al., 1991*). SNPs from the Illumina Immuno Bead-Chip (manifest file: Immuno_BeadChip_11419691.bpm from Illumina, San Diego, CA, USA) were also assessed by the Center for Public Health Genomics at the University of Virginia as described previously (*Krischer et al., 2017*).

## Study design

### Primary outcome definition

The primary TEDDY outcome variable was the development of persistent, confirmed IA. IA was assessed every 3 months through four years of age. Persistent autoimmunity was defined by the confirmed (two reference labs in agreement) presence of any one of three islet autoantibodies:IAA, GADA, or insulinoma antigen-2 (IA-2A) on two or more consecutive visits. Date of persistent autoimmunity was defined as the draw date of the first sample of the two consecutive autoantibody-positive samples.

### Islet autoantibodies

Islet autoantibodies were the first primary endpoint in the TEDDY study. Islet autoantibodies to insulin, GAD65, and IA-2 were measured in two laboratories by radiobinding assays (*Babaya et al., 2009*; *Bonifacio et al., 2010*). In the U.S., all sera were assayed at the Barbara Davis Center for Childhood Diabetes at the University of Colorado Denver; in Europe, all sera were assayed at the University of Bristol, the U.K. Assays in both laboratories have previously demonstrated high sensitivity, specificity, and concordance (*Törn et al., 2008*). All positive islet autoantibodies and 5% of negative samples were re-tested in the other reference laboratory and deemed confirmed if concordant. To optimize concordance, harmonized assays for GADA and IA-2A replaced earlier assays in January 2010 (Based on a receiver-operator curve analysis, prior samples that needed to be re-analyzed with the harmonized assays included: Denver GADA between –0.015 and 0.042; Bristol GADA between 10.69 and 36.72; Denver IA-2A between –0.004 and 0.016; and Bristol IA-2A between 6.69 and 10.58). To distinguish maternal antibodies from IA in the child, the IA status of the mother was measured when the child was aged 9 months, and the child's IA status was measured at 3 months of age and then every three months until 18 months of age. If the child had positive IA during this time, the status was listed as 'pending' until 18 months of age, at which time the child's IA status was determined based on both maternal and child IA over the first 18 months of the study. If maternal antibodies were present, the child was not considered persistently IA positive until positive at/after 18 months, unless the child had a negative sample prior to their first positive sample. A recognized limitation of this approach is that true IA positivity during the first 18 months of life that waned could have been masked by maternal antibodies.

### Diagnosis of T1D and CD

T1D and CD were the second primary endpoints in the TEDDY study. At the time of clinical diagnosis of diabetes, data were collected outside the TEDDY clinics (care providers or clinics) using a standardized case report form requiring documentation to fulfill American Diabetes Association criteria for classification of T1D (*Elding Larsson et al., 2014*).

TEDDY children were screened for tTGA using radiobinding assays from 2 years of age and annually thereafter. TEDDY samples from the US sites were analyzed at the Barbara Davis Center and samples from the European sites at the Bristol University, which was chosen as the reference laboratory as previously described (*Liu et al., 2014*). All samples from US children with a tTGA level >0.01 units were sent to the Bristol laboratory for analysis. Children with a positive tTGA had additional samples collected at three-month intervals to 48 months of age, and at 6-month intervals after the 48 month visit. In addition, all tTGA positive children had previously collected samples analyzed to find the closest time point to conversion. Children with a positive tTGA result in two consecutive samples were defined as having CDA (i.e. primary outcome) and referred to local health care providers for follow-up and clinical evaluation of CD. The decision to perform a diagnostic intestinal biopsy was

outside the TEDDY protocol, but it was recommended by the TEDDY investigators of Celiac Disease Committee to biopsy CDA children with tTGA levels 30 units or greater in CDA children with gastrointestinal symptoms regardless of tTGA level. Diagnosis of CD per se was defined as an intestinal biopsy showing a Marsh score >1, or if a biopsy was not performed, having a mean tTGA level of 100 units or greater in two consecutive samples (i.e. secondary outcome), respectively.

## Statistical and computational analyses

### Whole genome sequencing

Whole genome sequencing (WGS) was conducted on the subjects based on the IA and T1D nested case–control (NCC) studies with targeted 30× coverage by Macrogen, Inc, Rockville, MD, U.S.A as described previously (*Törn et al., 2022*). Joint variant discovery and genotype calling was performed by combining TEDDY and TOPMed WGS data using the topmed_variant_calling pipeline (https://github.com/statgen/topmed_variant_calling; *LeFaive and Min Kang, 2024*; *Taliun et al., 2021*).

### Determination of tri-SNP haplotype

Tri-SNP haplotypes were extracted from the ImmunoChip or WGS phased variant call file using scikit-allel software v1.3.5 (https://scikit-allel.readthedocs.io/en/stable/).

### Determination of B8-DR3 haplotype

Extended haplotype status of the individuals was imputed based on a subset of 203 SNPs whose non-reference alleles were associated with the B8-DR3 haplotype (*A*01:01-B*08:01-DRB1*03:01*) (*Gourraud et al., 2014*). The ImmunoChip variant calls contained 79 of these variants, whereas the WGS data contained 200. The individuals were assigned a B8-DR3 dosage based on the mode of the alternative allele counts of all SNPs, i.e., 0 if most SNPs were called homozygous reference genotype, 1 if most loci were heterozygous, and 2 if most loci were homozygous alternate allele.

### Genetic ancestry inference

Out of 7759 individuals, 5277 (68.0%) had self-reported race/ethnicity information as African American (65), Hispanic (558), and White non-Hispanic (4654). These were assigned to AFR, AMR, and EUR superpopulation groups, respectively. The remaining 2482 individuals' population membership was inferred using the software Kinship-based INference for Gwas (KING v2.2.7, *Manichaikul et al., 2010*) ancestry inference commands (`king -b reference.bed,samples.bed --pca --projection`). Briefly, the program uses 2409 samples from the 1000 Genomes Project with known population information to create Principal Components (PCs) and projects the study samples onto these PCs based on their genetic variation data (from ImmunoChip and Whole Genome Sequencing for TEDDY samples). Python sklearn library's (v0.24.2) RandomForestClassifier module was used to assign population membership using the PCs generated by KING. The classifier's performance was assessed by splitting the 2409 individuals from 1000 Genomes Project with known ancestry into training (70%) and test (30%) sets which showed an accuracy of 99.6%. For ancestry assignment of study samples, all 2409 samples from the 1000 Genomes Project were used as the training data and all 7903 TEDDY samples (all samples with genetic data, including those that were not in the final data set, e.g., with ineligible HLA types) as test data. Study samples where the ancestry information was reported were compared to the classifier predictions which showed 95.6% concordance. The inferred population information was used only for the 2482 samples that did not have the self-reported ancestry data.

### Statistical model

The Cox Proportional Hazards (PH) method was used to analyze the impact of tri-SNP haplotypes on primary and secondary T1D and CD disease outcomes. T1D diagnosis, islet autoimmunity (IA), first-appearing islet autoantibody, celiac diagnosis (CD), and celiac disease autoantibody (CDA) status were defined as events in separate models; age of the subject (in months) was used as the time component. When the event was not observed, age at the latest clinic visit (for CD or T1D) or at the last negative serum sample collection (for IA and CDA) was used as the right-censor time. In the first appearing autoantibody models, samples were right-censored if any other antibody than the one in the model outcome first appeared.

Known risk factors, including HLA type, sex, having a first-degree relative with the disease and genetic variants previously shown to be associated with each outcome (*Table 8*; *Krischer et al., 2017*; *Sharma et al., 2016*; *Sharma et al., 2018*), as well as potential confounders, including ancestry/ethnicity and the country of residence were included in the model to adjust for them. Tri-SNP 101 allele was modeled as a numerical variable (0, 1, or 2 alleles) representing an additive genetic effect, and hence, its hazard ratio values should be interpreted as HR per each additional allele. Similarly, the known GWAS loci were encoded numerically (0, 1, 2 alternate alleles). For the categorical covariates HLA type, sex, country, and genetic ancestry, the baselines were assigned to DR4/DR8, female, USA, and EUR, respectively. Cox PH modeling was carried out using the Lifelines Python library v0.26.3 (*Davidson-Pilon, 2019*).

## RNA sequence analysis

The RNA samples were prepared using Illumina's TruSeq Stranded mRNA Sample Prep Kit from the whole blood samples. RNA sequencing was conducted using the Illumina HiSeq4000 platform with paired-end 2×101 bp reads with a targeted 50 million reads per sample by the Broad Institute, Cambridge, MA. Gene expression was quantified using Salmon software v1.6.0 (*Patro et al., 2017*) and the Gencode v39 transcript set. Differential gene expression analysis was carried out using limma v3.48.3 (*Ritchie et al., 2015*) with the design formula '~batch + age + sex + triSNP 101' to account for differences due to sequencing batch, age, and the sex of each sample. Age was modeled as a categorical: younger or older than 1 year old as the major difference due to age was in children younger than 1 based on PCA analysis of gene expression data (not shown). Multiple samples from the same individual were accounted for by using the individual as the blocking factor as recommended in the limma user manual. Independent effect of triSNP-101 was extracted from the model using the *topTable* function and coefficient triSNP 101.

## Copy number analysis

A bed file containing 1 kilobase non-overlapping windows for MHC region (chromosome 6 positions 28510000–33482000) was created. MHC locus read coverage for each sample was extracted from WGS reads mapped to human genome assembly hg38 using samtools v1.16.1 bedcov command and the MHC bed file. '-Q 30' option was used for counting uniquely mapping reads. The average normalized coverage for C4A and C4B genes using uniquely mapping reads (*Figure 3*), as well as combined C4 average coverage using all reads (*Figure 3—figure supplement 1A*) were calculated.

## Total C4 GCN estimation

We estimated the total C4 GCN based on the read coverage of the C4 region, including C4A and C4B but excluding the intronic HERV insertion (*Figure 3—figure supplement 1A*). Histograms of per sample coverage for C4 region, as well as 25 kilobases flanking regions on both sides, were plotted (*Figure 3—figure supplement 1B*). While the flanking regions showed a single peak around 2, indicating a normal diploid status; C4 region showed three peaks around 1, 1.5, and 2, likely representing 2, 3, and 4 GCNs, respectively. Since the three peaks in the C4 histogram was well separated, we estimated total C4 GCN based on these peaks: samples under the first peak (coverage <1.4) were assigned 2 GCN; those under the second peak (1.4 < coverage < 1.9) were assigned 3 GCN and those under the third peak (coverage >1.9) were assigned 4 GCN.

## C4A and C4B GCN estimation

Histograms of coverage based on reads uniquely mapping to either C4A or C4B were created (*Figure 3—figure supplement 1C*). We observed a major peak at 0 coverage for C4A, indicating total C4A deletions and a separate peak around 0.3 indicating 1 GCN; and 2 peaks for C4B around 0.3 and 0.6, which likely indicate GCNs of 1 and 2. Since total gene deletions were well separated in the histograms for C4A and C4B, we first assigned the GCNs of C4A or C4B null samples (average coverage below 0.04). However, because the peaks for 1 and 2 GCNs were not clearly separated, the rest of the C4A and C4B GCNs were estimated based on the total C4 GCN and the ratio of unique coverage of C4A and C4B. The exact heuristic used was: if a sample was C4A null, C4B was assigned the same value as total C4 GCN; if C4B was null, C4A was assigned the C4 GCN. For samples where

neither C4A nor C4B was null: if C4 GCN was 2, C4A and C4B each were assigned 1 GCN; if C4 GCN was 3, C4A and C4B coverage were compared and larger received 2 GCN and smaller 1 GCN; if C4 GCN was 4, C4A and C4B were each assigned 2 GCN unless the coverage ratio between them was larger than 2.5, in which case the gene with more coverage was assigned 3 and the other 1 GCN.

## Acknowledgements

Full details for the TEDDY Study Group can be found in Appendix 1. The TEDDY Study is funded by U01 DK63829, U01 DK63861, U01 DK63821, U01 DK63865, U01 DK63863, U01 DK63836, U01 DK63790, UC4 DK63829, UC4 DK63861, UC4 DK63821, UC4 DK63865, UC4 DK63863, UC4 DK63836, UC4 DK95300, UC4 DK100238, UC4 DK106955, UC4 DK112243, UC4 DK117483, U01 DK124166, U01 DK128847, and Contract No. HHSN267200700014C from the National Institute of Diabetes and Digestive and Kidney Diseases (NIDDK), National Institute of Allergy and Infectious Diseases (NIAID), Eunice Kennedy Shriver National Institute of Child Health and Human Development (NICHD), National Institute of Environmental Health Sciences (NIEHS), Centers for Disease Control and Prevention (CDC), and JDRF. This work is supported in part by the NIH/NCATS Clinical and Translational Science Awards to the University of Florida (UL1 TR000064) and the University of Colorado (UL1 TR002535). The content is solely the responsibility of the authors and does not necessarily represent the official views of the National Institutes of Health.

## Additional information

### Funding

| Funder | Grant reference number | Author |
| --- | --- | --- |
| National Institute of Diabetes and Digestive and Kidney Diseases | U01 DK63861 | TEDDY Study Group |
| National Institute of Allergy and Infectious Diseases | | TEDDY Study Group |
| Eunice Kennedy Shriver National Institute of Child Health and Human Development | | TEDDY Study Group |
| National Institute of Environmental Health Sciences | | TEDDY Study Group |
| National Institute of Diabetes and Digestive and Kidney Diseases | U01 DK63829 | TEDDY Study Group |
| National Institute of Diabetes and Digestive and Kidney Diseases | U01 DK63821 | TEDDY Study Group |
| National Institute of Diabetes and Digestive and Kidney Diseases | U01 DK63865 | TEDDY Study Group |
| National Institute of Diabetes and Digestive and Kidney Diseases | U01 DK63863 | TEDDY Study Group |
| National Institute of Diabetes and Digestive and Kidney Diseases | U01 DK63836 | TEDDY Study Group |
| National Institute of Diabetes and Digestive and Kidney Diseases | U01 DK63790 | TEDDY Study Group |

| Funder | Grant reference number | Author |
| --- | --- | --- |
| National Institute of Diabetes and Digestive and Kidney Diseases | UC4 DK63829 | TEDDY Study Group |
| National Institute of Diabetes and Digestive and Kidney Diseases | UC4 DK63861 | TEDDY Study Group |
| National Institute of Diabetes and Digestive and Kidney Diseases | UC4 DK63821 | TEDDY Study Group |
| National Institute of Diabetes and Digestive and Kidney Diseases | UC4 DK63865 | TEDDY Study Group |
| National Institute of Diabetes and Digestive and Kidney Diseases | UC4 DK63863 | TEDDY Study Group |
| National Institute of Diabetes and Digestive and Kidney Diseases | UC4 DK63836 | TEDDY Study Group |
| National Institute of Diabetes and Digestive and Kidney Diseases | UC4 DK95300 | TEDDY Study Group |
| National Institute of Diabetes and Digestive and Kidney Diseases | UC4 DK100238 | TEDDY Study Group |
| National Institute of Diabetes and Digestive and Kidney Diseases | UC4 DK106955 | TEDDY Study Group |
| National Institute of Diabetes and Digestive and Kidney Diseases | UC4 DK117483 | TEDDY Study Group |
| National Institute of Diabetes and Digestive and Kidney Diseases | UC4 DK112243 | TEDDY Study Group |
| National Institute of Diabetes and Digestive and Kidney Diseases | U01 DK124166 | TEDDY Study Group |
| National Institute of Diabetes and Digestive and Kidney Diseases | U01 DK128847 | TEDDY Study Group |
| National Institute of Diabetes and Digestive and Kidney Diseases | HHSN267200700014C | TEDDY Study Group |

The funders had no role in study design, data collection and interpretation, or the decision to submit the work for publication.

## Author contributions

Ozkan Aydemir, Conceptualization, Data curation, Software, Formal analysis, Visualization, Writing – original draft, Writing – review and editing; Jeffrey A Bailey, Conceptualization, Resources, Formal analysis, Writing – original draft, Writing – review and editing; Daniel Agardh, Resources, Writing – original draft; Åke Lernmark, Conceptualization, Resources, Writing – original draft, Writing – review and editing; Janelle A Noble, Elizabeth P Blankenhorn, Conceptualization, Data curation, Writing – original draft, Writing – review and editing; Agnes Andersson Svärd, Anette-G Ziegler, Jorma Toppari, Beena Akolkar, William A Hagopian, Marian J Rewers, TEDDY Study Group, Resources; Hemang M Parikh, Data curation; John P Mordes, Conceptualization, Supervision, Investigation, Writing – original draft, Writing – review and editing

## Author ORCIDs
Ozkan Aydemir ⬤ https://orcid.org/0000-0002-3458-6527
Jeffrey A Bailey ⬤ https://orcid.org/0000-0002-6899-8204
Daniel Agardh ⬤ https://orcid.org/0000-0003-1435-1234
Åke Lernmark ⬤ https://orcid.org/0000-0003-1735-0499
Janelle A Noble ⬤ https://orcid.org/0009-0002-5529-1186
Agnes Andersson Svärd ⬤ https://orcid.org/0000-0002-6838-7382
Elizabeth P Blankenhorn ⬤ https://orcid.org/0000-0002-6331-0319
Hemang M Parikh ⬤ https://orcid.org/0000-0002-9076-6709
Anette-G Ziegler ⬤ https://orcid.org/0000-0002-6290-5548
William A Hagopian ⬤ https://orcid.org/0000-0003-2979-0475
Marian J Rewers ⬤ https://orcid.org/0000-0003-3829-9207
John P Mordes ⬤ https://orcid.org/0000-0003-2414-380X

## Ethics
Clinical trial registration NCT00279318.

Data were obtained from participants in the TEDDY prospective cohort study which has been described in detail (Teddy Study Group, 2007, 2008). The TEDDY study was approved by local US Institutional Review Boards and European Ethics Committee Boards, including the Colorado Multiple Institutional Review Board, Medical College of Georgia Human Assurance Committee (2004-2010), Georgia Health Sciences University Human Assurance Committee (2011-2012), Georgia Regents University Institutional Review Board (2013-2015), Augusta University Institutional Review Board (2015-present), University of Florida Health Center Institutional Review Board, Washington State Institutional Review Board (2004-2012), Western Institutional Review Board (2013-present), Ethics Committee of the Hospital District of Southwest Finland, Bayerische Landesärztekammer (Bavarian Medical Association) Ethics Committee, Regional Ethics Board in Lund, Section 2 (2004-2012), and Lund University Committee for Continuing Ethical Review (2013-present). All parents or guardians provided written informed consent before participation in genetic screening and enrollment. There was no compensation provided to participants. The analysis of the anonymized data provided by TEDDY to UMass was judged by UMass Institutional Review Board as an exempt study.

Reviewer #1 (Public Review): https://doi.org/10.7554/eLife.89068.2.sa1
Reviewer #2 (Public Review): https://doi.org/10.7554/eLife.89068.2.sa2

# Additional files

## Supplementary files
MDAR checklist

Supplementary file 1. Numerical results from all CoxPH analysis and differential gene expression analysis.

## Data availability
The code and source data for the analyses presented in this paper, including all numerical data used to generate each figure/table and all deidentified clinical data, are available and openly accessible at https://github.com/aydemiro/mp257 (copy archived at *Aydemir, 2025*). ImmunoChip, RNA-sequencing, and whole-genome sequencing data were generated as part of the The Environmental Determinants of Diabetes in the Young (TEDDY) study and submitted to the dbGaP repository (Study Accession: phs001442.v4.p3). Data from TEDDY study (https://doi.org/10.58020/y3jk-x087) reported here will be made available for request at the NIDDK Central Repository (NIDDK-CR) website, Resources for Research (R4R), https://repository.niddk.nih.gov/. The data are available under controlled access in accordance with dbGaP and NIDDK authorization to ensure that data are shared for type 1 diabetes and related disease research consistent with the TEDDY study participants' consent and protect participant privacy. Data access requests can be submitted directly to NIDDK-CR and dbGaP through their respective websites. These requests are reviewed by an assigned NIH Data Access Committee, and both repositories have eligibility requirements that investigators must meet to submit requests.

The following datasets were generated:

| Author(s) | Year | Dataset title | Dataset URL | Database and Identifier |
|---|---|---|---|---|
| TEDDY Study Group | 2008 | The Environmental Determinants of Diabetes in the Young Study (TEDDY) | https://www.ncbi.nlm.nih.gov/projects/gap/cgi-bin/study.cgi?study_id=phs001442.v4.p3 | NCBI dbGaP, phs001442.v4.p3 |
| Krischer J | 2025 | The Environmental Determinants of Diabetes in the Young (TEDDY) | https://doi.org/10.58020/y3jk-x087 | NIDDK Central Repository, 10.58020/y3jk-x087 |

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

## Appendix 1

## The TEDDY Study Group membership

### Colorado Clinical Center

Marian Rewers, M.D., Ph.D., PI[1,4,6,9,10], Kimberly Bautista[11], Judith Baxter[8,911], Daniel Felipe-Morales, Brigitte I. Frohnert, M.D., Ph.D.[2,13], Marisa Stahl, M.D.[12], Isabel Flores Garcia, Patricia Gesualdo[2,6,11,13], Sierra Hays, Michelle Hoffman[11,12,13], Rachel Karban[11], Edwin Liu, M.D.[12], Leila Loaiza Jill Norris, Ph.D.[2,3,11], Holly O'Donnell, Ph.D.[8], Loana Thorndahl, Andrea Steck, M.D.[3,13], Kathleen Waugh[6,7,11]

University of Colorado, Anschutz Medical Campus, Barbara Davis Center for Childhood Diabetes, Aurora, CO, United States.

### Finland Clinical Center

Jorma Toppari, M.D., Ph.D., PI[¥^1,4,10,13], Olli G. Simell, M.D., Ph.D., Annika Adamsson, Ph.D.[^11], Suvi Ahonen[*±§], Mari Åkerlund[*±§], Sirpa Anttila[μº], Leena Hakola[*±], Anne Hekkala, M.D.[μº], Tiia Honkanen[μº], Heikki Hyöty, M.D., Ph.D.[*±6], Jorma Ilonen, M.D., Ph.D.[¥3], Sanna Jokipuu[^], Taru Karjalainen[μº], Leena Karlsson[^], Jukka Kero M.D., Ph.D.[¥^3, 13], Jaakko J. Koskenniemi M.D., Ph.D.[¥^], Miia Kähönen[μº11,13], Mikael Knip, M.D., Ph.D.[*±], Minna-Liisa Koivikko[μº], Katja Kokkonen[*±], Merja Koskinen[*±], Mirva Koreasalo[*±§2], Kalle Kurppa, M.D., Ph.D.[*±12], Salla Kuusela, M.D. [μº], Jarita Kytölä[*±], Jutta Laiho, Ph.D.[6], Tiina Latva-aho[μº], Siiri Leisku[*±], Laura Leppänen[^], Katri Lindfors, Ph.D.[*12], Maria Lönnrot, M.D., Ph.D.[*±6], Elina Mäntymäki[^], Markus Mattila[*±], Maija Miettinen[§2], Teija Mykkänen[μº], Tiina Niininen[±*11], Sari Niinistö[§2], Noora Nurminen[*±], Sami Oikarinen, Ph.D.[*±6], Hanna-Leena Oinas[*±], Paula Ollikainen[μº], Zhian Othmani[¥], Sirpa Pohjola [μº], Solja Raja-Hanhela[μº], Jenna Rautanen[±§], Anne Riikonen[*±§2], Minna Romo[^], Juulia Rönkä[μº], Nelli Rönkä[μº], Satu Simell, M.D., Ph.D.[¥12], Päivi Tossavainen, M.D.[μº], Mari Vähä-Mäkilä[¥], Eeva Varjonen[^11], Riitta Veijola, M.D., Ph.D.[μº13], Irene Viinikangas[μº], Silja Vilmi[μº], Suvi M. Virtanen, M.D., Ph.D.[*±§2].

[¥]University of Turku, Turku, Finland
[*]Tampere University, Tampere, Finland
[μ]University of Oulu, Oulu, Finland
[^]Turku University Hospital, Hospital District of Southwest Finland, Turku, Finland
[±]Tampere University Hospital, Tampere, Finland
[º]Oulu University Hospital, Oulu, Finland
[§]Finnish Institute for Health and Welfare, Helsinki, Finland

### Georgia/Florida Clinical Center

Richard McIndoe, Ph.D., PI[^4,10], Desmond Schatz, M.D.[*4,7,8], Diane Hopkins[^11], Michael Haller, M.D.[*13], Risa Bernard[^11], Melissa Gardiner[^11], Ashok Sharma, Ph.D.[^], Laura Jacobsen, M.D.[*13], Jennifer Hosford[^], Kennedy Petty[^], Leah Myers[^], Chelsea Salmon[*]

[^]Center for Biotechnology and Genomic Medicine, Augusta University, Augusta, GA, United States
[*]University of Florida, Pediatric Endocrinology, Gainesville, FL, United States

### Germany Clinical Center

Anette G. Ziegler, M.D., PI[1,3,4,10], Ezio Bonifacio Ph.D.[*], Cigdem Gezginci, Willi Grätz, Anja Heublein, Eva Hohoff[¥2], Sandra Hummel, Ph.D.[2], Annette Knopff[7], Melanie Köger, Sibylle Koletzko, M.D.[¶12], Claudia Ramminger[11], Roswith Roth, Ph.D.[8], Jennifer Schmidt, Marlon Scholz, Joanna Stock[8,11,13], Katharina Warncke, M.D.[13], Lorena Wendel, Christiane Winkler, Ph.D.[2,11]

Forschergruppe Diabetes e.V. and Institute of Diabetes Research, Helmholtz Zentrum München, Forschergruppe Diabetes, and Klinikum rechts der Isar, Technische Universität München, Neuherberg, Germany

*Center for Regenerative Therapies, TU Dresden, Dresden, Germany

¶Dr. von Hauner Children's Hospital, Department of Gastroenterology, Ludwig Maximillians University Munich, Munich, Germany

¥University of Bonn, Department of Nutritional Epidemiology, Bonn, Germany

## Sweden Clinical Center

Åke Lernmark, Ph.D., PI[1,3,4,5,6,8,9,10], Daniel Agardh, M.D., Ph.D.[6,12], Carin Andrén Aronsson, Ph.D.[2,11,12], Rasmus Bennet, Corrado Cilio, Ph.D., M.D.[6], Susanne Dahlberg, Ulla Fält, Malin Goldman Tsubarah, Emelie Ericson-Hallström, Lina Fransson, Emina Halilovic, Gunilla Holmén, Susanne Hyberg, Berglind Jonsdottir, M.D., Ph.D.[11], Naghmeh Karimi, Helena Elding Larsson, M.D., Ph.D.[6,13], Marielle Lindström, Markus Lundgren, M.D., Ph.D.[13], Marlena Maziarz, Ph.D., Jessica Melin[11], Caroline Nilsson, Kobra Rahmati, Anita Ramelius, Falastin Salami, Ph.D., Anette Sjöberg, Evelyn Tekum Amboh Carina Törn, Ph.D.[3], Ulrika Ulvenhag, Terese Wiktorsson, Åsa Wimar[13]

Lund University, Lund, Sweden

## Washington Clinical Center

William A. Hagopian, M.D., Ph.D., PI[2,3,4,6,7,10,12,13], Michael Killian[6,7,11,12], Claire Cowen Crouch[11,13], Jennifer Skidmore[2], Trevor Bender, Megan Llewellyn, Cody McCall, Arlene Meyer, Jocelyn Meyer, Denise Mulenga[11], Nole Powell, Jared Radtke, Shreya Roy, Preston Tucker

Pacific Northwest Research Institute, Seattle, WA, United States

## Pennsylvania Satellite Center

Dorothy Becker, M.D., Margaret Franciscus, MaryEllen Dalmagro-Elias Smith[2], Ashi Daftary, M.D., Mary Beth Klein, Chrystal Yates

Children's Hospital of Pittsburgh of UPMC, Pittsburgh, PA, United States

## Data Coordinating Center

Jeffrey P. Krischer, Ph.D., PI[1,4,5,9,10], Rajesh Adusumali, Sarah Austin-Gonzalez, Maryouri Avendano, Sandra Baethke, Brant Burkhardt, Ph.D.[6], Martha Butterworth[2], Nicholas Cadigan, Joanna Clasen, Kevin Counts, Laura Gandolfo, Jennifer Garmeson, Veena Gowda, Christina Karges, Shu Liu, Xiang Liu, Ph.D.[2,3,8,13], Kristian Lynch, Ph.D.[6,8], Jamie Malloy, Lazarus Mramba, Ph.D.[2], Cristina McCarthy[11], Jose Moreno, Hemang M. Parikh, Ph.D.[3,8], Cassandra Remedios, Chris Shaffer, Susan Smith[11], Noah Sulman, Ph.D., Roy Tamura, Ph.D.[1,2,11,12,13], Dena Tewey, Henri Thuma, Michael Toth, Ulla Uusitalo, Ph.D.[2], Kendra Vehik, Ph.D.[4,5,6,8,13], Ponni Vijayakandipan, Melissa Wroble, Jimin Yang, Ph.D., R.D.[2], Kenneth Young, Ph.D

*Past staff: Michael Abbondondolo, Lori Ballard, Rasheedah Brown, David Cuthbertson, Stephen Dankyi, Christopher Eberhard, Steven Fiske, David Hadley, Ph.D., Kathleen Heyman, Belinda Hsiao, Francisco Perez Laras, Hye-Seung Lee, Ph.D., Qian Li, Ph.D., Colleen Maguire, Wendy McLeod, Aubrie Merrell, Steven Meulemans, Ryan Quigley, Laura Smith, Ph.D*

University of South Florida, Tampa, FL, United States

## Project scientist

Beena Akolkar, Ph.D.[1,3,4,5,6,7,9,10]

National Institutes of Diabetes and Digestive and Kidney Diseases, Bethesda, MD, United States

## Autoantibody Reference Laboratories

Liping Yu, M.D.[^,5], Dongmei Miao, M.D.[^], Kathlee Gillespie*[5], Kyla Chandler*, Ilana Kelland*, Yassin Ben Khoud*, Matthew Randell*

^Barbara Davis Center for Childhood Diabetes, University of Colorado Denver
*Bristol Medical School, University of Bristol, United Kingdom

## Genetics Laboratory

Stephen S. Rich, Ph.D.[3], Wei-Min Chen, Ph.D.[3], Suna OnengutGumuscu, Ph.D.[3], Emily Farber, Rebecca Roche Pickin, Ph.D., Jonathan Davis, Jordan Davis, Dan Gallo, Jessica Bonnie, Paul Campolieto
Center for Public Health Genomics, University of Virginia, Charlottesville, VA, United States

## HLA Reference Laboratory

William Hagopian[3], M.D., Ph.D., Jared Radtke, Preston Tucker
Pacific Northwest Research Institute, Seattle, WA, United States
*Previously Henry Erlich, Ph.D.[3], Steven J. Mack, Ph.D., Anna Lisa Fear. Center for Genetics, Children's Hospital Oakland Research Institute*

## Repository

Chris Deigan
NIDDK Biosample Repository at Fisher BioServices, Rockville, MD, United States
(Previously: Ricky Schrock, Polina Malone, Sandra Ke, Niveen Mulholland, Ph.D.)

## Other contributors

Thomas Briese, Ph.D.[6], Columbia University
Todd Brusko, Ph.D.[5], University of Florida, Gainesville, FL, United States
Teresa Buckner, Ph.D.[2], University of Northern Colorado, Greeley, CO, United States
Suzanne Bennett Johnson, Ph.D.[8,11], Florida State University, Tallahassee, FL, United States
Eoin McKinney, Ph.D.[5], University of Cambridge, Cambridge, United Kingdom
Tomi Pastinen, M.D., Ph.D.[5], The Children's Mercy Hospital, Kansas City, MO, United States
Steffen Ullitz Thorsen, M.D., Ph.D.[2], Department of Clinical Immunology, University of Copenhagen, Copenhagen, Denmark, and Department of Pediatrics and Adolescents, Copenhagen University Hospital, Herlev, Denmark
Eric Triplett, Ph.D.[6],University of Florida, Gainesville, FL, United States

## Committees

[1]Ancillary Studies, [2]Diet, [3]Genetics, [4]Human Subjects/Publicity/Publications, [5]Immune Markers, [6]Infectious Agents, [7]Laboratory Implementation, [8]Psychosocial, [9]Quality Assurance, [10]Steering, [11]Study Coordinators, [12]Celiac Disease, [13]Clinical Implementation.

