## [Editor Report · eLife assessment]

This study presents **valuable** findings on genetic risk factors for type 1 diabetes and celiac disease using a large cohort from the Environmental Determinants of Diabetes in the Young (TEDDY) study. The evidence supporting the claims of the authors is **solid**, although the inclusion of the genetic effect of this locus on individuals with different genetic backgrounds would have strengthened the study. The work will be of interest to population geneticists working on diabetes and celiac disease.

---

## [Referee Report · Reviewer #1 (Public Review)]

Polymorphisms in genes in the human leukocyte antigen (HLA) class II region comprise the most important inherited risk factors for many autoimmune diseases including type 1 diabetes (T1D) and celiac disease (CD). The paper focuses on the novel triad ((SNPs): rs3135394, rs9268645, and rs3129877) finding quite interesting results. The paper suggests further studies at the molecular and structural level to increase our fundamental knowledge of the etiology of autoimmune deceases.

---

## [Referee Report · Reviewer #2 (Public Review)]

In this manuscript, Aydemir et al. utilized the large TEDDY study and examined the effect of previously identified tri-SNP in the HLA-DRA gene on the risk of type 1 diabetes (T1D) and celiac disease (CD). They confirmed the protective effect of the tri-SNP haplotype "101" on T1D development. Meanwhile, the same haplotype appeared to be positively associated with risk for CD and the development of CD autoimmunity. The authors further explored the molecular effect of different tri-SNP haplotypes. They proposed that C4A and C4B might be the downstream target.

Overall, the study is rigorously conducted with proper statistical methods applied. The tri-SNP could be used as an additional risk factor when estimating T1D and celiac disease susceptibility in genetic screening. However, how this locus be incorporated into the current scheme of genetic screening is not discussed and is unlikely to be straightforward.